# Unfavorable Changes of Platelet Reactivity on Clopidogrel Therapy Assessed by Impedance Aggregometry Affect a Larger Volume of Acute Ischemic Lesions in Stroke

**DOI:** 10.3390/diagnostics11030405

**Published:** 2021-02-27

**Authors:** Adam Wiśniewski, Joanna Sikora, Aleksandra Karczmarska-Wódzka, Przemysław Sobczak, Adam Lemanowicz, Elżbieta Zawada, Rytis Masiliūnas, Dalius Jatužis

**Affiliations:** 1Department of Neurology, Faculty of Medicine, Nicolaus Copernicus University in Toruń, Collegium Medicum in Bydgoszcz, 85-094 Bydgoszcz, Poland; 2Biotechnology Research and Teaching Team, Department of Transplantology and General Surgery, Faculty of Medicine, Nicolaus Copernicus University in Toruń, Collegium Medicum in Bydgoszcz, 85-094 Bydgoszcz, Poland; joanna.sikora@cm.umk.pl (J.S.); akar@cm.umk.pl (A.K.-W.); przemyslawsobczak02@gmail.com (P.S.); 3Department of Radiology and Diagnostic Imaging, Faculty of Medicine, Nicolaus Copernicus University in Toruń, Collegium Medicum in Bydgoszcz, 85-094 Bydgoszcz, Poland; adam.lemanowicz@gmail.com (A.L.); e.zawada13@gmail.com (E.Z.); 4Center of Neurology, Vilnius University, LT-08661 Vilnius, Lithuania; rytis.masiliunas@santa.lt (R.M.); dalius.jatuzis@mf.vu.lt (D.J.)

**Keywords:** platelet reactivity, clopidogrel resistance, stroke, infarct volume, ischemic infarct, chronic vascular changes, platelet function

## Abstract

Background: High on-treatment platelet reactivity or its equivalent—resistance to the antiplatelet agent—significantly reduces the efficacy of the therapy, contributing to a negative impact on stroke course. Previous studies demonstrated that aspirin resistance is associated with a larger size of acute ischemic infarct. Due to the increasing use of clopidogrel in the secondary prevention of stroke, we aimed to assess the impact of clopidogrel resistance on the size and extent of ischemic lesions, both acute and chronic. Methods: This prospective, single-center and observational study involved 74 ischemic stroke subjects, treated with 75 mg of clopidogrel. We used impedance aggregometry to determine platelet reactivity 6–12 h after a dose of clopidogrel as a first assessment and 48 h later as the second measurement. A favorable dynamics of platelet reactivity over time was the decrease in the minimum value equal to the median in the entire study. The volume of acute ischemic infarct was estimated within 48 h after onset in diffusion-weighted imaging and fluid-attenuated inversion recovery sequences of magnetic resonance and the severity of chronic vascular lesions by Fazekas scale. Results: Subjects with mild severity of chronic vascular lesions (Fazekas 1) exhibited a significant decrease of platelet reactivity over time (*p* = 0.035). Dynamics of platelet reactivity over time differed between subjects with large, moderate, mild and insignificant size of acute ischemic lesion (Kruskall-Wallis H = 3.2576; *p* = 0.048). In multivariate regression models, we reported unfavorable dynamics of platelet reactivity alone and combined with a high initial value of platelet reactivity as independent predictors of higher risk of a significant ischemic infarct volume (OR 7.16 95%CI 1.69–30.31, *p* = 0.008 and 26.49 95%CI 1.88–372.4, *p* = 0.015, respectively). Conclusions: We emphasized that unfavorable dynamics of platelet reactivity over time during clopidogrel therapy in acute phase of stroke affect the volume of acute infarct and the severity of chronic vascular lesions.

## 1. Introduction

Clopidogrel plays a significant role in the secondary prevention after myocardial infarction. Its importance in antiplatelet therapy after ischemic stroke has also increased recently. Due to the high prevalence of aspirin resistance and high occurrence of recurrent ischemic stroke, it is increasingly used as a monotherapy in nonembolic ischemic stroke, instead of aspirin [1]. Dual antiplatelet therapy with the simultaneous use of clopidogrel and aspirin is becoming more common in minor strokes or high risk transient ischemic attacks [2,3,4]. Its efficacy and safety in the secondary prevention of ischemic stroke has been demonstrated and confirmed [5]. Similar to aspirin, platelet inhibition in some cases might be limited or variable. High on-clopidogrel platelet reactivity or clopidogrel resistance is a phenomenon that significantly reduces the efficacy of the antiaggregant, leading to a drug failure and decreased platelet inhibitory effect [6]. Some relevant papers showed its high prevalence and negative impact on clinical outcome and recovery, reflecting its significance in stroke therapy [7,8,9,10,11].

In contrast to aspirin, the effect of resistance on the size of an acute ischemic lesion and the severity of chronic ischemic changes in the brain has not been analyzed. Our previous study on aspirin has shown that this effect is correlated with stroke etiology, and is only relevant for stroke caused by large-vessel disease [12]. We hypothesize that a similar relationship with the volume of ischemic focus may exist depending on the effectiveness of clopidogrel treatment, which could be assessed in platelet function testing [13]. However, a single time measurement of platelet reactivity might be insufficient in appropriate evaluation of high on-clopidogrel platelet reactivity, due to its variability [14]. Therefore, we improved our research and performed a double assessment of platelet reactivity, assuming that the changes of platelet reactivity over time would better reflect the role of the phenomenon being analyzed than just a single measurement.

The aim of the current study was to evaluate the impact of high on-treatment platelet reactivity and changes of platelet reactivity over time, during clopidogrel therapy, on the size of the acute ischemic focus, as well as on the severity of chronic ischemic changes in the brain.

## 2. Materials and Methods

### 2.1. Study Design and Participants

This prospective, single-center and observational study was conducted from November 2019 to December 2020 in a stroke intensive care unit in the Department of Neurology at the University Hospital No. 1 in Bydgoszcz, Poland. We enrolled 74 subjects who met both the clinical and radiological criteria for the recognition of ischemic stroke [15]. According to the guidelines, all stroke subjects received 150 mg of aspirin immediately after excluding hemorrhagic stroke based on computed tomography on admission [1]. After this period, from the second day of stroke, we switched to clopidogrel (75 mg a day) for the secondary prevention of stroke. Due to the lack of strict and evidence-based recommendations for timing of the switch, we decided on fast implementation of clopidogrel after a minimal period dedicated to aspirin-only treatment. The priority was to assess dynamics of platelets in the acute phase of stroke. We included participants with large-vessel disease (at least 50% stenosis of artery corresponding to the stroke symptoms) or small-vessel disease (typical morphological changes in the neuroimaging). Sample size was calculated based on the prevalence of ischemic stroke in the general population of the province (Bydgoszcz, Poland) using the available sample size calculator. The recommended (estimated) sample size was 69 subjects with a confidence level of 90% based on a population of 400,000.

The following exclusion criteria were used: the duration of stroke symptoms over 24 h before enrollment, subjects underwent reperfusion therapy (intravenous thrombolysis and/or endovascular treatment), inability to sign informed consent (e.g., consciousness disturbances or moderate/severe aphasia), cardioembolic etiology of stroke (documented atrial fibrillation, dilated cardiomyopathy, thrombus in the heart cavities or new diagnosed atrial fibrillation during hospitalization), contraindications to magnetic resonance imaging (e.g., pacemaker or severe claustrophobia), a history of stroke or transient ischemic attack (TIA) in the previous 3 years, documented neoplasms with higher risk of bleeding, taking antiplatelet agents or low molecular weight heparin before stroke, a history of a significant bleeding in the previous 2 years, level of hemoglobin <9 g/dL and thrombocytopenia < 100,000/µL. We adopted the following criteria for cardiovascular risk factors: hypertension (recognized before enrollment or blood pressure values over 140/80 mmHg), diabetes (recognized before enrollment or fasting glucose level of over 200 mg/dL), hyperlipidemia (recognized before enrollment or values of cholesterol in serum over 180 mg/dL), smoking (current smoking), obesity (body mass index over 30) and alcohol abuse (at least two beers or 100 mL of vodka more than 15 days per month for a period of 3 months).

### 2.2. Platelet Reactivity Testing

The platelet function testing was assessed by impedance aggregometry in the Laboratory of Experimental Biotechnology at Collegium Medicum in Bydgoszcz. Blood samples were collected 6–12 h after intaking the first dose of clopidogrel, as the first time-point. The second measurement of platelet reactivity was performed 48 h later (±4 h). The Multiplate–Dynabyte multichannel platelet function analyzer (Roche Diagnostics, France) was used in this study, applied as an adenosine diphosphate (ADP) test, where ADP was an agonist of the platelets. Addition of platelet activator to the solution forces the platelets to move towards two electrodes, which translates into changes in resistance (impedance). The device automatically converts these signals into area under the curve (AUC) units, and the average for two electrode pairs was reported as a final result of the measurement. The cut-off values above 46 AUC were considered as high on-treatment platelet reactivity, denoting non-effective inhibition of platelets by clopidogrel. The determination of such a limit value complies with the manufacturer’s recommendations and was often used in other research [16,17]. Individual processing steps were identical to those reported in other studies [18]. A comparison of the values obtained in both measurements determined the changes of platelet reactivity. We assumed as favorable changes a decrease in platelet reactivity values between two assessments of at least the median difference obtained in the entire study (at least 5 AUC). All different changes were considered unfavorable. Samples are shown in Figure 1. The significance of changes of platelet reactivity was based on statistical calculations.

### 2.3. The Volumetric Evaluation

A 1.5 tesla Optima 450w scanner (G.E. Healthcare, Chicago, IL, USA) in in the Department of Radiology at the University Hospital No. 1 in Bydgoszcz was used to perform magnetic resonance imaging within the first 48 h from the onset of stroke symptoms. Diffusion-weighted images were quantitatively analyzed using a standard diagnostic software package (Functool 4.4, Advantage Workstation 4.4, G.E. Healthcare, Chicago, IL, USA). The steps for determining the volume of ischemic areas in the DWI and FLAIR images were shown in Figure 2. In every series of DWI and FLAIR images, appropriate threshold values of the signal intensities were set, so that only voxels overlapping with the ischemic area remained in the image. Other voxels within the selected intensity range but located outside the region of interest (e.g., noise) were then manually removed from all layers. In the last step, a dedicated function was used to automatically calculate the volume of displayed voxels [19]. The final result of the volume was provided in cm^3^. On the basis of volumetric assessment, we subdivided the subjects into strokes with significant and insignificant acute ischemic size, setting a volume of 2 cm^3^ as the limit value. Next, the group with a significant focus in DWI was further divided into subjects with large (>15 cm^3^), moderate (>5 cm^3^) and small (2–5 cm^3^) stroke volume. The volume in DWI was also used for logistic regression analysis. We based our assessment of the severity of chronic ischemic lesions on the Fazekas scale [20].

### 2.4. Ethical Statement

The study protocol received a positive opinion of the Bioethics Committee of the Nicolaus Copernicus University in Torun at Collegium Medicum of Ludwik Rydygier in Bydgoszcz (KB number 735/2019 on 29.10.2019). All subjects before enrollment read the study protocol and signed informed consent to participate in the study. The study was conducted according to the Declaration of Helsinki.

### 2.5. Statistical Evaluation Methods

The statistical analysis was performed with STATISTICA, version 13.1 (Dell company, TX, USA). The collected data were presented as median and range. Non-parametric tests were used, i.e., Mann–Whitney U test (assessment of changes of platelet reactivity over time), Spearman’s rank correlation test (correlation between clopidogrel resistance and acute or chronic brain lesions). Univariate and multivariate logistic regression models were assessed for evaluation of predictive properties of high on-treatment platelet reactivity. The level of *p* < 0.05 was considered as the threshold for statistical significance.

### 2.6. Definitions of Clopidogrel Resistance

For logistic regression models, we introduced three different definitions of clopidogrel resistance. We based the first definition only on the initial value of platelet reactivity and we set results above 46 AUC as our definition of resistance. We based the second definition only on the changes of platelet reactivity over time (difference between two assessments) and set all cases of unfavorable changes (the recorded increase or decrease lower than the median—5 AUC) as a resistance. Definition 3 included the fulfillment of both conditions adopted for definition 1 and definition 2 jointly.

## 3. Results

### 3.1. Overall Results

The baseline characteristics of the participants included to the study are shown in Table 1. In the first time point, the median platelet reactivity was 51 AUC (range 13–107 AUC), while in the second time point, 46 AUC (9–103 AUC), which means a median decrease in platelet activity over time at 5 AUC. The subgroup with large-vessel etiology of stroke had significantly larger volume of ischemic focus than a small-vessel disease subgroup (DWI median 5.53 vs 1.78 cm^3^, *p* < 0.0001; FLAIR median 5.89 vs 1.99 cm^3^, *p* < 0.0001).

### 3.2. Platelet Reactivity and Chronic Ischemic Lesions

In the subgroup with mild severity of chronic vascular lesions (FAZEKAS – 1), we reported a significant and favorable decrease in platelet reactivity over time (median 52.5 AUC vs 46 AUC, *p* = 0.035), whereas in subgroups with moderate (FAZEKAS – 2) and extensive (FAZEKAS – 3) severity of chronic vascular lesions, no significant or favorable decrease in platelet reactivity was noted (median 50 AUC vs 47.5 AUC, *p* = 0.529 and 33.5 AUC vs 29 AUC, *p* = 0.810, respectively) (Figure 3). No significant correlations between platelet reactivity values in the first and the second assessment and the severity of chronic brain lesions were observed (R = −0.189, *p* = 0.101 and R = −0.119, *p* = 0.325, respectively).

### 3.3. Platelet Reactivity and Acute Ischemic Lesions

We found significant correlations between acute ischemic infarct volume in DWI and platelet reactivity values, both in the first measurement (R = 0.253, *p* = 0.029) and in the second measurement (R = 0.612, *p* < 0.0001), shown in Figure 4, as well as in FLAIR (R = 0.254, *p* = 0.029 and R = 0.613, *p* < 0.0001, respectively).

In the subgroup with insignificant size of acute ischemic infarct (in DWI), we reported significant and favorable changes of platelet reactivity over time (*p* = 0.027), whereas no significant and favorable changes were observed in the subgroup with significant volume of acute ischemic lesion (*p* = 0.528). Similar dependencies were noted in the FLAIR sequence (*p* = 0.039 and *p* = 0.512, respectively). Breaking down a group with a significant focus into individual subgroups, we revealed that changes of platelet reactivity over time differed significantly when comparing subjects with large, moderate, small and insignificant volume of acute ischemic lesion (Kruskall-Wallis H = 3.258; *p* = 0.048, median AUC 75 vs 81.5, 54 vs 61, 50 vs 49 and 49.5 vs 37, respectively) (Figure 5). Notably, the subjects with a large and moderate size of acute ischemic infarct exhibited an extremely unfavorable increase in platelet reactivity over time.

### 3.4. Logistic Regression Models

Clopidogrel resistance in the first definition did not affect the risk of the occurrence of the significant size of acute ischemic infarct (OR 1.67 95%CI 0.63–4.38, *p* = 0.299). Clopidogrel resistance in the second and the third definition was associated with higher risk of a significant size of acute ischemic infarct (OR 3.62 95%CI 1.24–10.6, *p* = 0.019 and OR 11.61 95%CI 1.41–94.91, *p* = 0.022, respectively). The risk of the individual significant sizes of infarct (large, moderate, small) compared to the insignificant infarct volume in clopidogrel-resistant subjects according to the selected definitions is presented in Table 2.

Due to insignificant dependencies in Definition 1 in the univariate logistic regression analysis, only the second and the third definition were taken for further analysis. We developed two multivariate logistic regression models adjusted for age, sex, etiology of stroke, common risk factors for vascular diseases and clopidogrel resistance based on the adopted second and third definition. In both models, we reported that clopidogrel resistance and large-vessel etiology of stroke were independent predictors of a higher risk of a significant ischemic infarct volume (Table 3).

## 4. Discussion

To the best of our knowledge, this study is the first attempt to assess the impact of clopidogrel high on-treatment platelet reactivity on the size of acute ischemic infarct. The strengths of this research are the evaluation of the impact on chronic ischemic changes and undoubtedly the double assessment of platelet function over time, which made it possible to track the impact of the dynamics of changes in platelet reactivity on the size of the acute focus or the severity of chronic brain lesions.

We are the first to highlight that clopidogrel-resistant stroke subjects are more likely to develop a larger size of acute ischemic infarct. Our findings are consistent with similar studies regarding the impact of aspirin resistance on the volume of acute ischemic lesion [21,22]. We hypothesize that the reason for this state lies in similar mechanisms underlying the higher platelet reactivity during antiplatelet therapy [6]. However, data on the role of aspirin resistance in extension of ischemic size remain ambiguous and contradictory, as some studies showed no significant dependencies in this field [23,24]. In our opinion, this might be due to the fact that a single measurement of platelet activity at a single time point is not sufficient to properly assess its effect. In the current study, we revealed a similar relationship with clopidogrel. Although the high platelet reactivity, both during the first and second measurement, still correlated with the size of the ischemic lesion, in logistic regression, it did not significantly increase the probability of a more extensive focus. Therefore, it seems more appropriate to use the assessment of the changes in platelet reactivity over time for this purpose, as the difference between two measurements. Favorable changes in the dynamics of platelet reactivity, expressed as a significant decrease in values over time, were associated with the limited volume of acute infarct. Furthermore, along with the increase in the volume of the ischemic focus in the brain, the dynamics of platelet reactivity over time became increasingly unfavorable. Additionally, we have shown that the unfavorable dynamics of platelet reactivity is an independent predictor influencing the size and extent of acute ischemic lesions. It cannot be ruled out that the discrepancies in the studies on the effect of aspirin resistance on the size of the infarct may have been due to the lack of analysis of the changes of platelet reactivity over time, which, as we have reported for clopidogrel, corresponds better with the extent of ischemic lesions than a single measurement.

We hypothesize that the optimal solution for the proper evaluation of the actual impact of platelet reactivity on the size of the focus is the simultaneous assessment of the value of platelet reactivity and their changes over time. Similarly to the dynamics measured alone, the combination of high platelet reactivity values and more unfavorable changes over time were also associated with an increase in the volume of acute infarct, statistically even more significant than the dynamics itself. Unlike our previous aspirin study, which showed a role of stroke etiology in significant correlations between platelet reactivity and infarct size, we did not find a similar relationship in the current clopidogrel study [12]. Nevertheless, we did not previously analyze the dynamics of changes in platelet reactivity over time, which may explain the reported discrepancies and the independence of the influence of clopidogrel resistance on the extent of ischemic focus from the cause of stroke.

A novel finding reported in this study is also an impact of a significant and favorable decrease in platelet reactivity over time on the prevalence of mild severity chronic vascular changes in the brain. We did not observe any favorable dynamics of platelet reactivity at higher levels of severity of chronic vascular lesions or significant correlations between the values of platelet reactivity and the severity of chronic vascular lesions. Our results are inconsistent with data reported by Lundstroem et al., the only study that we found on this topic [25]. They revealed that clopidogrel resistance (measured also by impedance aggregometry) is associated with more extensive chronic vascular lesions. However, the determination of platelet function was 1 month after the ischemic event, they included only subjects after minor stroke and assessed chronic changes on the basis of computed tomography scans. Our findings suggest that sequential determination of platelet reactivity over time has an advantage over a single measurement, and further research in this direction should focus on their dynamics rather than individual values.

This study contains several limitations. This research was based on the small sample size that requires verification among larger cohorts. For formal reasons, the study excluded patients with severe stroke, with impaired consciousness or severe speech disorders, who are unable to give informed consent, which would have made a full cross-sectional assessment impossible. Unfortunately, there is still no standardization in measuring platelet function. Therefore, the results obtained with one device should be approached carefully.

## 5. Conclusions

In summary, we emphasized the significance of the phenomenon of high on-clopidogrel platelet reactivity and the independent impact of unfavorable changes in its dynamics on the size and extent of acute ischemic infarct as well as the severity of chronic vascular changes. We conclude that appropriate platelet inhibitory effect of clopidogrel, and especially the maintenance of its beneficial impact over a period of time, is crucial for the effectiveness of treatment and is associated with a lower risk of extensive acute and chronic ischemic changes. Therefore, our findings support the sequential determination of platelet function in the acute phase of ischemic stroke not only to estimate the efficacy of the antiplatelet agent, but above all as an important prognostic factor related to the clinical outcome.

## Figures and Tables

**Figure 1 diagnostics-11-00405-f001:**
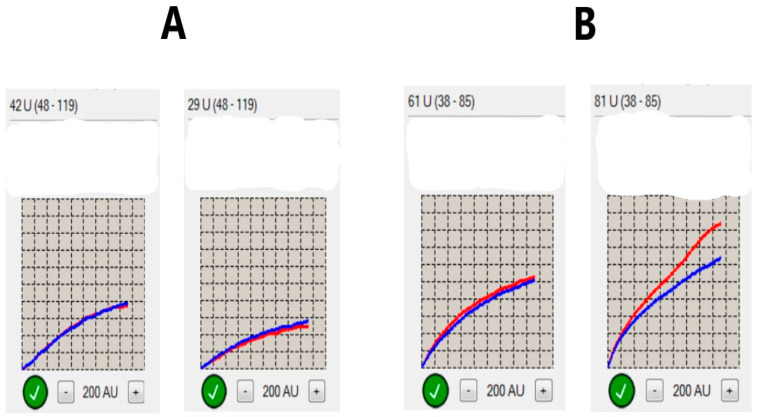
Changes of platelet reactivity over time. (**A**) A sample of favorable changes in platelet reactivity. On the left side—normal initial platelet reactivity value (42 U—area under the curve units). On the right side—48 h later—decrease in platelet reactivity to 29 U; the difference between the two assessments is 13 U—more than a median difference (5U). (**B**) A sample of unfavorable changes in platelet reactivity. On the left side—high initial platelet reactivity value (61 U). On the right side—48 h later; an increase in platelet reactivity to 81 U. This sample meets our criteria for the third definition of clopidogrel resistance (high initial value combined with unfavorable dynamics). The final value in AUC is the average between the results obtained in two pairs of electrodes (marked as red and blue lines).

**Figure 2 diagnostics-11-00405-f002:**
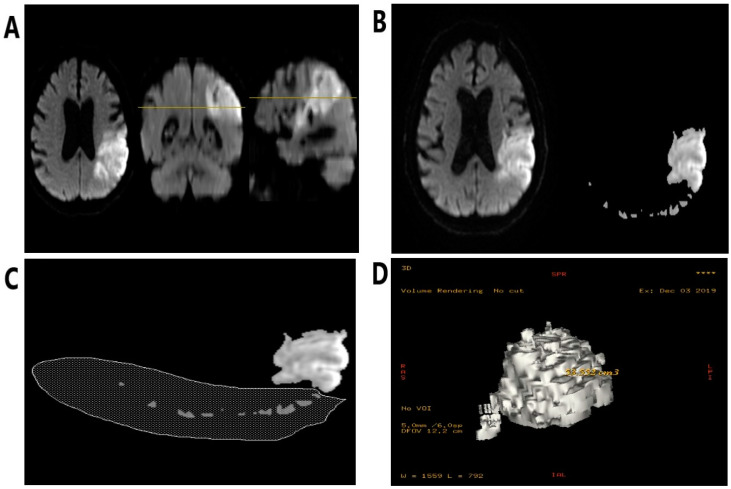
The steps for determining the volume of acute ischemic infarct. (**A**) Primary magnetic resonance (MR) image from a diffusion-weighted imaging (DWI) sequence; native axial view, and additional coronal and sagittal reconstructions. (**B**) On the left side, the original axial MR image from the DWI sequence. On the right side, the effect of limiting the displayed voxels to a chosen signal intensity range. (**C**) Manual removal of residual voxels (noise) outside of the ischemic stroke area. (**D**) Volumetric reformation of stroke with its volume displayed.

**Figure 3 diagnostics-11-00405-f003:**
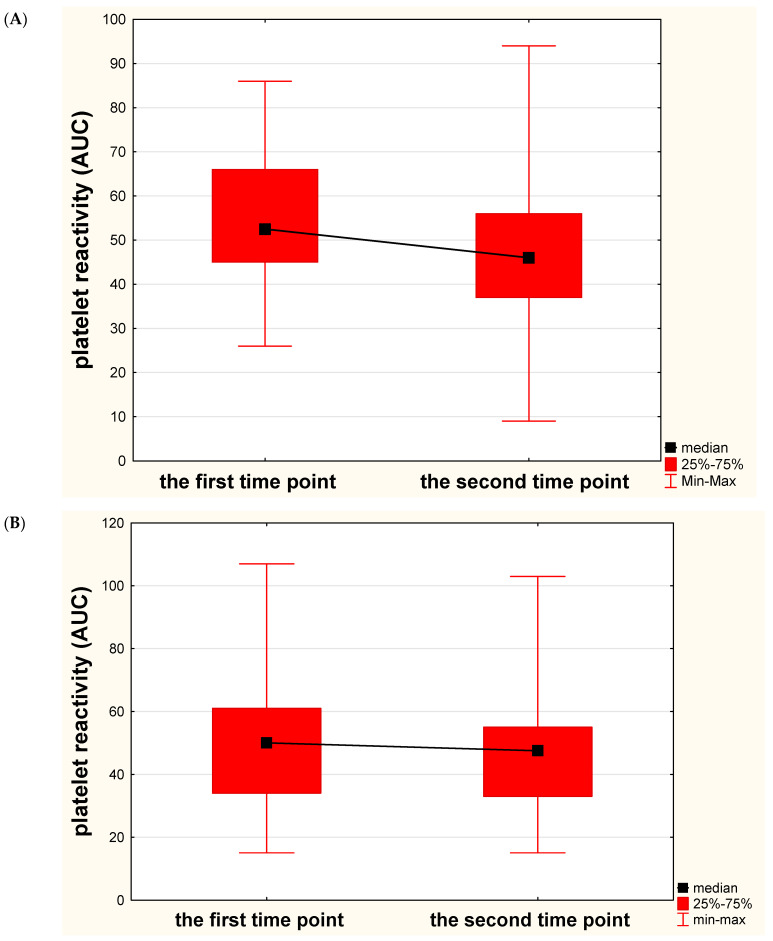
Changes of platelet reactivity over time in relation to the severity of chronic ischemic lesions. (**A**) Significant and favorable changes of platelet reactivity (expressed in area under the curve units—AUC) in stroke subjects with mild severity of chronic ischemic lesions (Fazekas scale—1). (**B**) Insignificant and unfavorable changes of platelet reactivity in stroke subjects with moderate severity of chronic ischemic lesions (Fazekas scale—2). (**C**) Insignificant and unfavorable changes of platelet reactivity in stroke subjects with extensive severity of chronic ischemic lesions (Fazekas scale—3).

**Figure 4 diagnostics-11-00405-f004:**
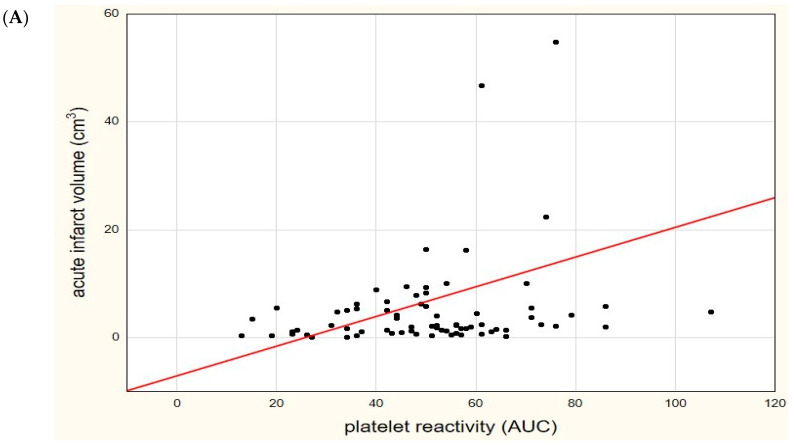
Significant correlation between platelet reactivity (in area under the curve units—AUC) and a volume of acute ischemic infarct (in cm^3^) in the diffusion-weighted sequence of magnetic resonance imaging. (**A**) Initial (the first time point) platelet reactivity value, (**B**) control (the second time point) platelet reactivity value.

**Figure 5 diagnostics-11-00405-f005:**
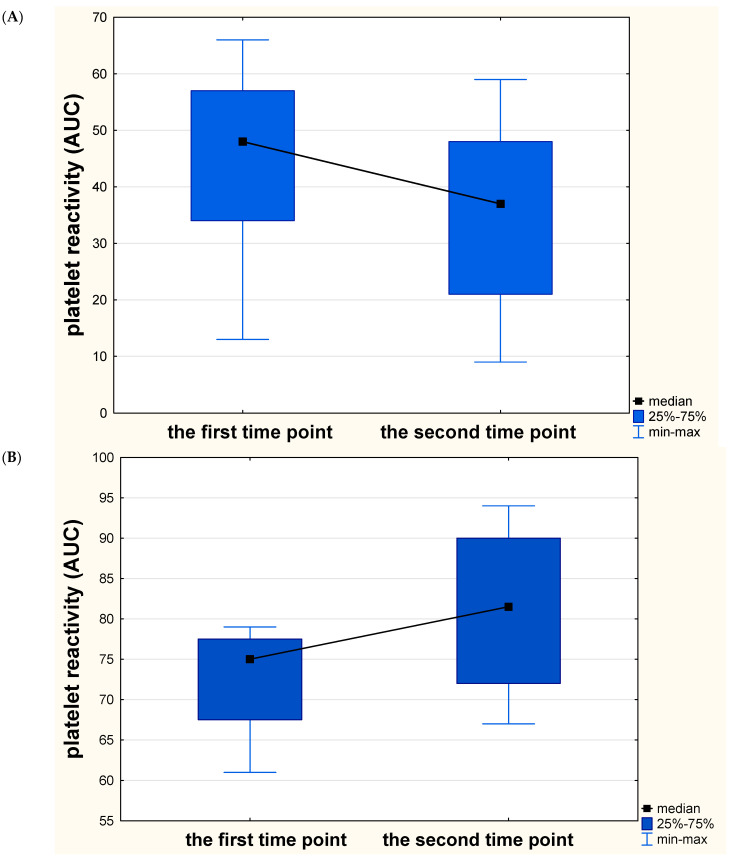
Dynamics of changes in platelet reactivity over time in relation to the size of acute ischemic infarct. Significant and favorable changes of platelet reactivity (expressed in area under the curve units—AUC) in stroke subjects with insignificant (<2 cm^3^) size of acute ischemic infarct in the diffusion-weighted sequence of magnetic resonance imaging (**A**). Insignificant and unfavorable changes of platelet reactivity in stroke subjects with large (>15 cm^3^) (**B**), moderate (5–15 cm^3^) (**C**) and small (2–5 cm^3^) (**D**) size of acute ischemic infarct. Extremely unfavorable increase of platelet reactivity can be seen among subjects with a large and moderate size of acute ischemic infarct.

**Table 1 diagnostics-11-00405-t001:** The general characteristics of ischemic stroke subjects (*n* = 74).

Parameter	Values
Age	67.5 (18–91)
Sex:	
Male	36 (48.6%)
Female	38 (51.4%)
Hypertension	53 (71.6%)
Diabetes	19 (25.7%)
Hyperlipidemia	24 (32.4%)
Smoking	22 (29.7%)
Obesity	22 (29.7%)
Alcohol abuse	6 (8.1%)
CRP (mg/L)	2.37 (0.21–148.96)
HbA1c (%)	5.8 (4.9–13.01)
D-Dimer (mg/mL)	451 (165–5926)
Fibrinogen (mg/dL)	335 (212–658)
Platelet count (thousands/µL)	256 (112–545)
NIHSS on admission	3 (1–16)
mRS on admission	2 (0–5)
The volume of ischemic focus DWI (cm^3^)	2.3 (0.09–136.9)
The volume of ischemic focus FLAIR (cm^3^)	2.5 (0.13–143.3)
Resistance to clopidogrel	47 (63.5%)
Etiology of stroke:	
Large vessel disease	18 (24.3%)
Small vessel disease	56 (75.4%)

FLAIR—fluid-attenuated inversion recovery; DWI—diffusion-weighted imaging; CRP—C-reactive protein; HbA1c—glycated hemoglobin, mRS—modified Rankin Scale, NIHSS—the National Institutes of Health Stroke Scale. Values for age, biochemical findings, scales and infarct volume are expressed as median and range. Values for sex, risk factors, clopidogrel resistance and stroke etiology are expressed as *n* and as a percentage.

**Table 2 diagnostics-11-00405-t002:** Univariate logistic regression of the risk of selected volumes of acute ischemic infarcts among clopidogrel-resistant versus -sensitive subjects, depending on three different definitions of the resistance.

	Definition1	Definition 2	Definition 3
	OR (95%CI) *p*	OR (95%CI) *p*	OR (95%CI) *p*
Infarct volume (vs insignificant) Mild	1.20 (0.44, 3.26) 0.724	6.5 (0.88, 47.9) 0.066	6.89 (0.78, 60.88) 0.083
Moderate	2.74 (0.27, 27.4) 0.391	3.19 (1.04, 9.84) 0.043 *	46.5 (3.2, 676.21) 0.005 *
Large	6.23 (0.30, 125.5) 0.234	13.0 (1.14, 147.8) 0.039 *	93.0 (4.56, 1895.27) 0.003 *

*—significant dependencies, OR—odds ratio, CI—confidence interval.

**Table 3 diagnostics-11-00405-t003:** Multivariate logistic regression analysis of predictors of a significant size of acute ischemic infarct in two models, depending on the definition of the resistance.

	Model 1 (Definition 2)		Model 2 (Definition 3)	
	Adjusted OR (95% CI)	*p*	Adjusted OR (95% CI)	*p*
Age	1.06 (0.99, 1.13)	0.076	1.07 (0.99, 1.14)	0.05
Clopidogrel resistance	7.16 (1.69, 30.31)	0.008 *	26.49 (1.88, 372.44)	0.015 *
Sex (male)	0.45 (0.12, 1.7)	0.241	0.57 (0.16, 2.09)	0.398
Diabetes	1.01 (0.23, 4.51)	0.985	1.94 (0.41, 9.09)	0.402
Large vessel disease	23.99 (2.39, 241.4)	0.007 *	28.46 (2.74, 295.47)	0.005 *
Smoking	0.23 (0.05, 0.99)	0.054	0.25 (0.06, 1.05)	0.059
Hyperlipidemia	1.41 (0.36, 5.51)	0.623	0.99 (0.24, 4.03)	0.988
Hypertension	0.97 (0.12, 4.84)	0.966	1.35 (0.27, 6.8)	0.715
Obesity	0.36 (0.07, 1.69)	0.193	0.31 (0.06, 1.52)	0.149

*—significant dependencies, OR—odds ratio, CI—confidence interval.

## Data Availability

The data that support the findings of this study are available from the corresponding author upon request.

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
