# Peer review of "Unfavorable Changes of Platelet Reactivity on Clopidogrel Therapy Assessed by Impedance Aggregometry Affect a Larger Volume of Acute Ischemic Lesions in Stroke"

_diagnostics, 2021, doi:10.3390/diagnostics11030405_

Round 1
Reviewer 1 Report
The authors presented interesting study Clopidogrel treatment in stroke patients.
The Introduction section is well written with adequate interpretation and explanation of the importance of clopidogrel treatment as antiplatelet therapy in ischemic stroke.
The hypotheses and aims are well postulated.
In methods section the study design was well documented and exclusion criteria were precisely pointed. However, the study sample should be stressed how it was gathered despite the statement in limitations section that the sample size is small. The sample size probably would fit the question, or the flow chart of patients.
The platelet reactivity testing and volumetric evaluations are well described, as well as definitions of clopidogrel resistance.
In Results section the figures are understandable and adequate textual interpretation was given.
The logistic regression models in tables are adequate but I would suggest that p values should be on 3 decimals for example: 0.076 not 0.0763. For the p values same applies when interpreting the figures in results.
The Discussion section is well written with adequate citation of representative literature when discussing the obtained results. Authors also highlighted the strengths of the study as well as limitations.
The Conclusions reflect the postulated aims, however due to the small sample size they are somehow to sound.
Reviewer 2 Report
The study is of great scientific interests. The authors designed and described the study properly. They also discussed the results in a good way showing the strengths of the study. The authors are also aware of the limitations of the study.
I have only minor issues to consider:
- Have the authors performed a power analysis to confirm that the numbers of participants provide sufficient power?
- I think Table 1 should be transferred into Results section. Also, consider if Figures 1 and 2 would be better in the description of Results.
- Results section should be structured. Divide it into subsections for greater clarity.
- The resolution of all Figures should be improved as captions within figures are not visible properly.
Reviewer 3 Report
This study reported the impact of high on-treatment platelet reactivity after 6-12 hours and 48 hours after taking clopidogrel. Authors found that clopidogrel resistant stroke subjects are more likely to develop a larger size of acute ischemic stroke.
Article is interesting but there are some methodological aspects that need to be clarified.
- Report definitions of diseases (dyslipidemia, hypertension etc)
- Are there patients under LWMH before admitting to the Hospital?
- How many patients were taking lipid lowering drugs? In particular, there are some statins that interfere with clopidogrel
- Any data about the timing of symptoms onset and the prescription of aspirin?
- I suggest to include timing of MR from symptoms onset in the analyses
